# Online Optimization for Max-Norm Regularization

**Jie Shen**
Dept. of Computer Science
Rutgers University
Piscataway, NJ 08854
js2007@rutgers.edu

**Huan Xu**
Dept. of Mech. Engineering
National Univ. of Singapore
Singapore 117575
mpexuh@nus.edu.sg

**Ping Li**
Dept. of Statistics
Dept. of Computer Science
Rutgers University
pingli@stat.rutgers.edu

## Abstract

Max-norm regularizer has been extensively studied in the last decade as it promotes an effective low rank estimation of the underlying data. However, max-norm regularized problems are typically formulated and solved in a batch manner, which prevents it from processing big data due to possible memory bottleneck. In this paper, we propose an online algorithm for solving max-norm regularized problems that is scalable to large problems. Particularly, we consider the matrix decomposition problem as an example, although our analysis can also be applied in other problems such as matrix completion. The key technique in our algorithm is to reformulate the max-norm into a matrix factorization form, consisting of a basis component and a coefficients one. In this way, we can solve the optimal basis and coefficients alternatively. We prove that the basis produced by our algorithm converges to a stationary point asymptotically. Experiments demonstrate encouraging results for the effectiveness and robustness of our algorithm.
*See the full paper at arXiv:1406.3190.*

## 1 Introduction

In the last decade, estimating low rank matrices has attracted increasing attention in the machine learning community owing to its successful applications in a wide range of domains including subspace clustering [13], collaborative filtering [9] and visual texture analysis [25], to name a few. Suppose that we are given an observed data matrix $Z$ of size $p \times n$, *i.e.*, $n$ observations in $p$ ambient dimensions, with each observation being i.i.d. sampled from some unknown distribution, we aim to learn a prediction matrix $X$ with a low rank structure to approximate $Z$. This problem, together with its many variants, typically involves minimizing a weighted combination of the residual error and matrix rank regularization term.

Generally speaking, it is intractable to optimize a matrix rank [15]. To tackle this challenge, researchers suggest alternative convex relaxations to the matrix rank. The two most widely used convex surrogates are the nuclear norm [1] [15] and the max-norm [2] [19]. In the work of [6], Candès *et al.* proved that under mild conditions, solving a convex optimization problem consisting of a nuclear norm regularization and a weighted $\ell_1$ norm penalty can exactly recover the low-rank component of the underlying data even if a constant fraction of the entries are arbitrarily corrupted. In [20], Srebro and Shraibman studied collaborative filtering and proved that the max-norm regularization formulation enjoyed a lower generalization error than the nuclear norm. Moreover, the max-norm was shown to empirically outperform the nuclear norm in certain practical applications as well [11, 12].

To optimize a max-norm regularized problem, however, algorithms proposed in prior work [12, 16, 19] require to access all the data. In a large scale setting, the applicability of such batch optimiza-

tion methods will be hindered by the memory bottleneck. In this paper, by utilizing the matrix factorization form of the max-norm, we propose an online algorithm to solve max-norm regularized problems. The main advantage of online algorithms is that the memory cost is independent from the sample size, which makes online algorithms a good fit for the *big data* era [14, 18].

Specifically, we are interested in the max-norm regularized matrix decomposition (MRMD) problem. Assume that the observed data matrix $Z$ can be decomposed into a low rank component $X$ and a sparse one $E$, we aim to simultaneously and accurately estimate the two components, by solving the following convex program:

$$\min_{X,E} \frac{1}{2} \|Z - X - E\|_F^2 + \frac{\lambda_1}{2} \|X\|_{\max}^2 + \lambda_2 \|E\|_{1,1}. \tag{1.1}$$

Here $\|\cdot\|_F$ denotes the Frobenius norm, $\|\cdot\|_{\max}$ is the max-norm (which promotes low rank), $\|\cdot\|_{1,1}$ is the $\ell_1$ norm of a matrix seen as a vector, and $\lambda_1$ and $\lambda_2$ are two non-negative parameters.

Our **main contributions** are two-folds: 1) We develop an online method to solve this MRMD problems, making it scalable to big data. 2) We prove that the solutions produced by our algorithm converge to a stationary point asymptotically.

## 1.1 Connection to Matrix Completion

While we mainly focus on the matrix decomposition problem, our method can be extended to the matrix completion (MC) problem [4, 7] with max-norm regularization [5], which is another popular topic in machine learning and signal processing. The MC problem can be described as follows:

$$\min_{X} \frac{1}{2} \|\mathcal{P}_\Omega (Z - X)\|_F^2 + \frac{\lambda}{2} \|X\|_{\max}^2,$$

where $\Omega$ is the set of indices of observed entries in $Z$ and $\mathcal{P}_\Omega(M)$ is the orthogonal projector onto the span of matrices vanishing outside of $\Omega$ so that the $(i,j)$-th entry of $\mathcal{P}_\Omega(M)$ is equal to $M_{ij}$ if $(i,j) \in \Omega$ and zero otherwise. Interestingly, the max-norm regularized MC problem can be cast into our framework. To see this, let us introduce an auxiliary matrix $M$, with $M_{ij} = C > 0$ if $(i,j) \in \Omega$ and $M_{ij} = \frac{1}{C}$ otherwise. The following reformulated MC problem,

$$\min_{X,E} \frac{1}{2} \|Z - X - E\|_F^2 + \frac{\lambda}{2} \|X\|_{\max}^2 + \|M \circ E\|_{1,1},$$

where "$\circ$" denotes the entry-wise product, is equivalent to our MRMD formulation (1.1). Furthermore, when $C$ tends to infinity, the reformulated problem converges to the original MC problem.

## 1.2 Related Work

Here we discuss some relevant work in the literature. Most previous works on max-norm focused on showing that the max-norm was empirically superior to the nuclear norm in a wide range of applications, such as collaborative filtering [19] and clustering [11]. Moreover, in [17], Salakhutdinov and Srebro studied the influence of data distribution for the max-norm regularization and observed good performance even when the data were sampled non-uniformly.

There are also studies which investigated the connection between the max-norm and the nuclear norm. A comprehensive study on this problem, in the context of collaborative filtering, can be found in [20], which established and compared the generalization bounds for the nuclear norm regularization and max-norm regularization, and showed that the generalization bound of the max-norm regularization scheme is superior. More recently, Foygel *et al.* [9] attempted to unify the nuclear norm and max-norm for gaining further insights on these two important regularization schemes.

There are few works to develop efficient algorithms for solving max-norm regularized problems, particularly large scale ones. Rennie and Srebro [16] devised a gradient-based optimization method and empirically showed promising results on large collaborative filtering datasets. In [12], the authors presented large scale optimization methods for max-norm constrained and max-norm regularized problems with a theoretical guarantee to a stationary point. Nevertheless, all those methods were formulated in a batch manner, which can be hindered by the memory bottleneck.

From a high level, the goal of this paper is similar to that of [8]. Motivated by the celebrated Robust Principal Component Analysis (RPCA) problem [6, 23, 24], the authors of [8] developed an online implementation for the nuclear-norm regularized matrix decomposition. Yet, since the max-norm is a much more complicated mathematical entity (*e.g.,* even the subgradient of the max-norm is not completely characterized to the best of our knowledge), new techniques and insights are needed in order to develop online methods for the max-norm regularization. For example, after taking the max-norm with its matrix factorization form, the data are still coupled and we propose to convert the problem to a constrained one for stochastic optimization.

The main technical contribution of this paper is to convert max-norm regularization to an appropriate matrix factorization problem amenable to online implementation. Part of our proof ideas are inspired by [14], which also studied online matrix factorization. In contrast to [14], our formulation contains an additive sparse noise matrix, which enjoys the benefit of robustness to sparse contamination. Our proof techniques are also different. For example, to prove the convergence of the dictionary and to well define their problem, [14] needs to assume that the magnitude of the learned dictionary is constrained. In contrast, in our setup we prove that the optimal basis is uniformly bounded, and hence our problem is naturally well defined.

## 2  Problem Setup

We first introduce our notations. We use bold letters to denote vectors. The $i$-th row and $j$-th column of a matrix $M$ are denoted by $\mathbf{m}(i)$ and $\mathbf{m}_j$, respectively. The $\ell_1$ norm and $\ell_2$ norm of a vector $\mathbf{v}$ are denoted by $\|\mathbf{v}\|_1$ and $\|\mathbf{v}\|_2$, respectively. The $\ell_{2,\infty}$ norm of a matrix is defined as the maximum $\ell_2$ row norm. Finally, the trace of a square matrix $M$ is denoted as $\mathrm{Tr}(M)$.

We are interested in developing an online algorithm for the MRMD Problem (1.1). By taking the matrix factorization form of the max-norm [19]:

$$\|X\|_{\max} \triangleq \min_{L,R}\{\|L\|_{2,\infty} \cdot \|R\|_{2,\infty} : \ X = LR^\top, L \in \mathbb{R}^{p \times d}, R \in \mathbb{R}^{n \times d}\}, \qquad (2.1)$$

where $d$ is the intrinsic dimension of the underlying data, we can rewrite Problem (1.1) into the following equivalent form:

$$\min_{L,R,E} \frac{1}{2}\|Z - LR^T - E\|_F^2 + \frac{\lambda_1}{2}\|L\|_{2,\infty}^2 \|R\|_{2,\infty}^2 + \lambda_2\|E\|_{1,1}. \qquad (2.2)$$

Intuitively, the variable $L$ corresponds to a basis and the variable $R$ is a coefficients matrix with each row corresponding to the coefficients. At a first sight, the problem can only be optimized in a batch manner as the term $\|R\|_{2,\infty}^2$ couples all the samples. In other words, to compute the optimal coefficients of the $i$-th sample, we are required to compute the subgradient of $\|R\|_{2,\infty}$, which needs to access all the data. Fortunately, we have the following proposition that alleviates the inter-dependency among samples.

**Proposition 2.1.** *Problem* (2.2) *is equivalent to the following constrained program:*

$$\begin{aligned} \underset{L,R,E}{\text{minimize}} \quad & \frac{1}{2}\|Z - LR^T - E\|_F^2 + \frac{\lambda_1}{2}\|L\|_{2,\infty}^2 + \lambda_2\|E\|_{1,1}, \\ \text{subject to} \quad & \|R\|_{2,\infty}^2 = 1. \end{aligned} \qquad (2.3)$$

Proposition 2.1 states that our primal MRMD problem can be transformed to an equivalent constrained one. In the new formulation (2.3), the coefficients of *each individual* sample (*i.e.*, a row of the matrix $R$) is *uniformly* constrained. Thus, the samples are decoupled. Consequently, we can, equipped with Proposition 2.1, rewrite the original problem in an online fashion, with each sample being separately processed:

$$\begin{aligned} \underset{L,R,E}{\text{minimize}} \quad & \frac{1}{2}\sum_{i=1}^{n}\|\mathbf{z}_i - L\mathbf{r}_i - \mathbf{e}_i\|_2^2 + \frac{\lambda_1}{2}\|L\|_{2,\infty}^2 + \lambda_2\sum_{i=1}^{n}\|\mathbf{e}_i\|_1, \\ \text{subject to} \quad & \forall i \in 1, 2, \dots, n, \|\mathbf{r}_i\|_2^2 \leq 1, \end{aligned}$$

where $\mathbf{z}_i$ is the $i$-th observed sample, $\mathbf{r}_i$ is the coefficients and $\mathbf{e}_i$ is the sparse error. Combining the first and third terms in the above equation, we have

$$\underset{L,R,E}{\text{minimize}} \quad \sum_{i=1}^{n} \tilde{\ell}(\mathbf{z}_i, L, \mathbf{r}_i, \mathbf{e}_i) + \frac{\lambda_1}{2}\|L\|_{2,\infty}^2, \tag{2.4}$$
$$\text{subject to} \quad \forall i \in 1, 2, \ldots, n, \|\mathbf{r}_i\|_2^2 \leq 1,$$

where

$$\tilde{\ell}(\mathbf{z}, L, \mathbf{r}, \mathbf{e}) \triangleq \frac{1}{2}\|\mathbf{z} - L\mathbf{r} - \mathbf{e}\|_2^2 + \lambda_2\|\mathbf{e}\|_1. \tag{2.5}$$

This is indeed equivalent to optimizing (*i.e.*, minimizing) the empirical loss function:

$$f_n(L) \triangleq \frac{1}{n}\sum_{i=1}^{n} \ell(\mathbf{z}_i, L) + \frac{\lambda_1}{2n}\|L\|_{2,\infty}^2, \tag{2.6}$$

where

$$\ell(\mathbf{z}, L) = \min_{\mathbf{r},\mathbf{e},\|\mathbf{r}\|_2^2 \leq 1} \tilde{\ell}(\mathbf{z}, L, \mathbf{r}, \mathbf{e}). \tag{2.7}$$

When $n$ goes to infinity, the empirical loss converges to the expected loss, defined as follows

$$f(L) = \lim_{n \to +\infty} f_n(L) = \mathbb{E}_{\mathbf{z}}[\ell(\mathbf{z}, L)]. \tag{2.8}$$

# 3 Algorithm

We now present our online implementation to solve the MRMD problem. The detailed algorithm is listed in Algorithm 1. Here we first briefly explain the underlying intuition: We optimize the coefficients $\mathbf{r}$, the sparse noise $\mathbf{e}$ and the basis $L$ in an alternating manner, which is known to be a successful strategy [8, 10, 14]. At the $t$-th iteration, given the basis $L_{t-1}$, we can optimize over $\mathbf{r}$ and $\mathbf{e}$ by examining the Karush Kuhn Tucker (KKT) conditions. To update the basis $L_t$, we then optimize the following objective function:

$$g_t(L) \triangleq \frac{1}{t}\sum_{i=1}^{t} \tilde{\ell}(\mathbf{z}_i, L, \mathbf{r}_i, \mathbf{e}_i) + \frac{\lambda_1}{2t}\|L\|_{2,\infty}^2, \tag{3.1}$$

where $\{\mathbf{r}_i\}_{i=1}^{t}$ and $\{\mathbf{e}_i\}_{i=1}^{t}$ have been computed in previous iterations. It is easy to verify that Eq. (3.1) is a surrogate function of the empirical cost function $f_t(L)$ defined in Eq. (2.6). The basis $L_t$ can be optimized by block coordinate decent, with $L_{t-1}$ being a warm start for efficiency.

# 4 Main Theoretical Results and Proof Outline

In this section we present our main theoretic result regarding the validity of the proposed algorithm. We first discuss some necessary assumptions.

## 4.1 Assumptions

1. The observed data are i.i.d. generated from a distribution with compact support $\mathcal{Z}$.

2. The surrogate functions $g_t(L)$ in Eq. (3.1) are strongly convex. Particularly, we assume that the smallest eigenvalue of the positive semi-definite matrix $\frac{1}{t}A_t$ defined in Algorithm 1 is not smaller than some positive constant $\beta_1$. Note that we can easily enforce this assumption by adding a term $\frac{\beta_1}{2}\|L\|_F^2$ to $g_t(L)$.

3. The minimizer for Problem (2.7) is unique. Notice that $\tilde{\ell}(\mathbf{z}, L, \mathbf{r}, \mathbf{e})$ is strongly convex w.r.t. $\mathbf{e}$ and convex w.r.t. $\mathbf{r}$. Hence, we can easily enforce this assumption by adding a term $\gamma\|\mathbf{r}\|_2^2$, where $\gamma$ is a small positive constant.

## 4.2 Main Theorem

The following theorem is the main theoretical result of this work. It states that when $t$ tends to infinity, the basis $L_t$ produced by Algorithm 1 converges to a stationary point.

**Theorem 4.1** (Convergence to a stationary point of $L_t$)**.** *Assume 1, 2 and 3. Given that the intrinsic dimension of the underlying data is $d$, the optimal basis $L_t$ produced by Algorithm 1 asymptotically converges to a stationary point of Problem* (2.8) *when $t$ tends to infinity.*

---

**Algorithm 1** Online Max-Norm Regularized Matrix Decomposition

---

**Input:** $Z \in \mathbb{R}^{p \times n}$ (observed samples), parameters $\lambda_1$ and $\lambda_2$, $L_0 \in \mathbb{R}^{p \times d}$ (initial basis), zero matrices $A_0 \in \mathbb{R}^{d \times d}$ and $B_0 \in \mathbb{R}^{p \times d}$

**Output:** optimal basis $L_t$

1: **for** $t = 1$ to $n$ **do**
2:     Access the $t$-th sample $\mathbf{z}_t$.
3:     Compute the coefficient and noise:

$$\{\mathbf{r}_t, \mathbf{e}_t\} = \underset{\mathbf{r}, \mathbf{e}, \|\mathbf{r}\|_2^2 \leq 1}{\arg \min} \ \tilde{\ell}(\mathbf{z}_t, L_{t-1}, \mathbf{r}, \mathbf{e}). \tag{3.2}$$

4:     Compute the accumulation matrices $A_t$ and $B_t$:

$$
\begin{aligned}
A_t &\leftarrow A_{t-1} + \mathbf{r}_t \mathbf{r}_t^\top, \\
B_t &\leftarrow B_{t-1} + (\mathbf{z}_t - \mathbf{e}_t)\, \mathbf{r}_t^\top.
\end{aligned}
$$

5:     Compute the basis $L_t$ by optimizing the surrogate function (3.1):

$$
\begin{aligned}
L_t &= \underset{L}{\arg \min} \ \frac{1}{t} \sum_{i=1}^{t} \tilde{\ell}(\mathbf{z}_i, L, \mathbf{r}_i, \mathbf{e}_i) + \frac{\lambda_1}{2t} \|L\|_{2,\infty}^2 \\
&= \underset{L}{\arg \min} \ \frac{1}{t} \left( \frac{1}{2} \operatorname{Tr}\left(L^\top L A_t\right) - \operatorname{Tr}\left(L^\top B_t\right) \right) + \frac{\lambda_1}{2t} \|L\|_{2,\infty}^2.
\end{aligned} \tag{3.3}
$$

6: **end for**

---

## 4.3 Proof Outline for Theorem 4.1

The essential tools for our analysis are from stochastic approximation [3] and asymptotic statistics [21]. There are three main steps in our proof:

**(I)** We show that the positive stochastic process $g_t(L_t)$ defined in Eq. (3.1) converges almost surely.

**(II)** Then we prove that the empirical loss function, $f_t(L_t)$ defined in Eq. (2.6) converges almost surely to the same limit of its surrogate $g_t(L_t)$. According to the central limit theorem, we can expect that $f_t(L_t)$ also converges almost surely to the expected loss $f(L_t)$ defined in Eq. (2.8), implying that $g_t(L_t)$ and $f(L_t)$ converge to the same limit.

**(III)** Finally, by taking a simple Taylor expansion, it justifies that the gradient of $f(L)$ taking at $L_t$ vanishes as $t$ tends to infinity, which concludes Theorem 4.1.

**Theorem 4.2** (Convergence of the surrogate function $g_t(L_t)$). *The surrogate function $g_t(L_t)$ we defined in Eq.* (3.1) *converges almost surely, where $L_t$ is the solution produced by Algorithm 1.*

To establish the convergence of $g_t(L_t)$, we verify that $g_t(L_t)$ is a quasi-martingale [3] that converges almost surely. To this end, we show that the expectation of the difference of $g_{t+1}(L_{t+1})$ and $g_t(L_t)$ can be upper bounded by a family of functions $\ell(\cdot, L)$ indexed by $L \in \mathcal{L}$, where $\mathcal{L}$ is a compact set. Then we show that the family of functions satisfy the hypotheses in the corollary of Donsker Theorem [21] and thus can be uniformly upper bounded. Therefore, we conclude that $g_t(L_t)$ is a quasi-martingale and converges almost surely.

Now let us verify the hypotheses in the corollary of Donsker Theorem. First we prove that the index set $\mathcal{L}$ is uniformly bounded.

**Proposition 4.3.** *Let $\mathbf{r}_t$, $\mathbf{e}_t$ and $L_t$ be the optimal solutions produced by Algorithm 1. Then,*

    *1. The optimal solutions $\mathbf{r}_t$ and $\mathbf{e}_t$ are uniformly bounded.*

    *2. The matrices $\frac{1}{t} A_t$ and $\frac{1}{t} B_t$ are uniformly bounded.*

3. *There exists a compact set $\mathcal{L}$, such that for all $L_t$ produced by Algorithm 1, $L_t \in \mathcal{L}$. That is, there exists a positive constant $L_{\max}$ that is uniform over $t$, such that for all $t > 0$,*

$$\|L_t\| \le L_{\max}.$$

To prove the third claim (which is required for our proof of convergence of $g_t(L_t)$), we should prove that for all $t > 0$, $\mathbf{r}_t$, $\mathbf{e}_t$, $\frac{1}{t}A_t$ and $\frac{1}{t}B_t$ can be uniformly bounded, which can easily be verified. Then, by utilizing the first order optimal condition of Problem (3.3), we can build an equation that connects $L_t$ with the four items we mentioned in the first and second claim. From Assumption 2, we know that the nuclear norm of $\frac{1}{t}A_t$ can be uniformly lower bounded. This property provides us the way to show that $L_t$ can be uniformly upper bounded. Note that in [8, 14], both papers assumed that the dictionary (or basis) is uniformly bounded. In contrast, here in the third claim of Proposition 4.3, we prove that such condition naturally holds in our problem.

Next, we show that the family of functions $\ell(\mathbf{z}, L)$ is uniformly Lipschitz w.r.t. $L$.

**Proposition 4.4.** *Let $L \in \mathcal{L}$ and denote the minimizer of $\ell(\mathbf{z}, L, \mathbf{r}, \mathbf{e})$ defined in (2.7) as:*

$$\{\mathbf{r}^*, \mathbf{e}^*\} = \arg\min_{\mathbf{r}, \mathbf{e}, \|\mathbf{r}\|_2 \le 1} \frac{1}{2}\|\mathbf{z} - L\mathbf{r} - \mathbf{e}\|_2^2 + \lambda_2\|\mathbf{e}\|_1.$$

*Then, the function $\ell(\mathbf{z}, L)$ defined in Problem (2.7) is continuously differentiable and*

$$\nabla_L \ell(\mathbf{z}, L) = (L\mathbf{r}^* + \mathbf{e}^* - \mathbf{z})\mathbf{r}^{*T}.$$

*Furthermore, $\ell(\mathbf{z}, \cdot)$ is uniformly Lipschitz and bounded.*

By utilizing the corollary of Theorem 4.1 from [2], we can verify the differentiability of $\ell(\mathbf{z}, L)$ and the form of its gradient. As all of the items in the gradient are uniformly bounded (Assumption 1 and Proposition 4.3), we show that $\ell(\mathbf{z}, L)$ is uniformly Lipschitz and bounded.

Based on Proposition 4.3 and 4.4, we verify that all the hypotheses in the corollary of Donsker Theorem [21] are satisfied. This implies the convergence of $g_t(L_t)$. We now move to step **(II)**.

**Theorem 4.5** (Convergence of $f(L_t)$). *Let $f(L_t)$ be the expected loss function defined in Eq. (2.8) and $L_t$ is the solution produced by the Algorithm 1. Then,*

1. *$g_t(L_t) - f_t(L_t)$ converges almost surely to 0.*

2. *$f_t(L_t)$ defined in Eq. (2.6) converges almost surely.*

3. *$f(L_t)$ converges almost surely to the same limit of $f_t(L_t)$.*

We apply Lemma 8 from [14] to prove the first claim. We denote the difference of $g_t(L_t)$ and $f_t(L_t)$ by $b_t$. First we show that $b_t$ is uniformly Lipschitz. Then we show that the difference between $L_{t+1}$ and $L_t$ is $O(\frac{1}{t})$, making $b_{t+1} - b_t$ be uniformly upper bounded by $O(\frac{1}{t})$. Finally, we verify the convergence of the summation of the serial $\{\frac{1}{t}b_t\}_{t=1}^\infty$. Thus, Lemma 8 from [14] applies.

**Proposition 4.6.** *Let $\{L_t\}$ be the basis sequence produced by the Algorithm 1. Then,*

$$\|L_{t+1} - L_t\|_F = O(\frac{1}{t}). \tag{4.1}$$

Proposition 4.6 can be proved by combining the strong convexity of $g_t(L)$ (Assumption 2 in Section 4.1) and the Lipschitz of $g_t(L)$; see the full paper for details.

Equipped with Proposition 4.6, we can verify that the difference of the sequence $b_t = g_t(L_t) - f_t(L_t)$ can be upper bounded by $O(\frac{1}{t})$. The convergence of the summation of the serial $\{\frac{1}{t}b_t\}_{t=1}^\infty$ can be examined by the expectation convergence property of quasi-martingale $g_t(L_t)$, stated in [3]. Applying the Lemma 8 from [14], we conclude that $g_t(L_t) - f_t(L_t)$ converges to zero $a.s.$.

After the first claim of Theorem 4.5 being proved, the second claim follows immediately, as $g_t(L_t)$ converges $a.s.$ (Theorem 4.2). By the central limit theorem, the third claim can be verified.

According to Theorem 4.5, we can see that $g_t(L_t)$ and $f(L_t)$ converge to the same limit $a.s.$ Let $t$ tends to infinity, as $L_t$ is uniformly bounded (Proposition 4.3), the term $\frac{\lambda_1}{2t}\|L_t\|_{2,\infty}^2$ in $g_t(L_t)$ vanishes. Thus $g_t(L_t)$ becomes differentiable. On the other hand, we have the following proposition about the gradient of $f(L)$.

**Proposition 4.7** (Gradient of $f(L)$)**.** *Let $f(L)$ be the expected loss function defined in Eq. (2.8). Then, $f(L)$ is continuously differentiable and $\nabla f(L) = E_{\mathbf{z}}[\nabla_L \ell(\mathbf{z}, L)]$. Moreover, $\nabla f(L)$ is uniformly Lipschitz on $\mathcal{L}$.*

Thus, taking a first order Taylor expansion for $f(L_t)$ and $g_t(L_t)$, we can show that the gradient of $f(L_t)$ equals to that of $g_t(L_t)$ when $t$ tends to infinity. Since $L_t$ is the minimizer for $g_t(L)$, we know that the gradient of $f(L_t)$ vanishes. Therefore, we have proved Theorem 4.1.

## 5 Experiments

In this section, we report some simulation results on synthetic data to demonstrate the effectiveness and robustness of our online max-norm regularized matrix decomposition (OMRMD) algorithm.

**Data Generation.** The simulation data are generated by following a similar procedure in [6]. The clean data matrix $X$ is produced by $X = UV^T$, where $U \in \mathbb{R}^{p \times d}$ and $V \in \mathbb{R}^{n \times d}$. The entries of $U$ and $V$ are i.i.d. sampled from the Gaussian distribution $\mathcal{N}(0, 1)$. We introduce a parameter $\rho$ to control the sparsity of the corruption matrix $E$, *i.e.*, a $\rho$-fraction of the entries are non-zero and following an i.i.d. uniform distribution over $[-1000, 1000]$. Finally, the observation matrix $Z$ is produced by $Z = X + E$.

**Evaluation Metric.** Our goal is to estimate the correct subspace for the underlying data. Here, we evaluate the fitness of our estimated subspace basis $L$ and the ground truth basis $U$ by the Expressed Variance (EV) [22]:

$$\text{EV}(U, L) \triangleq \frac{\text{Tr}(L^T U U^T L)}{\text{Tr}(UU^T)}.$$

The values of EV range in $[0, 1]$ and a higher EV value indicates a more accurate subspace recovery.

**Other Settings.** Through the experiments, we set the ambient dimension $p = 400$ and the total number of samples $n = 5000$ unless otherwise specified. We fix the tunable parameter $\lambda_1 = \lambda_2 = 1/\sqrt{p}$, and use default parameters for all baseline algorithms we compare with. Each experiment is repeated 10 times and we report the averaged EV as the result.

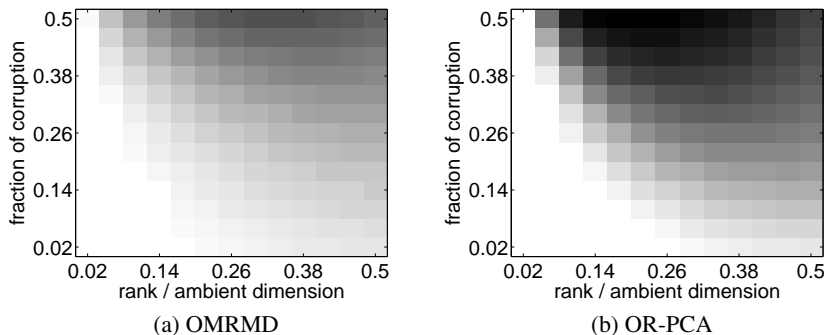

(a) OMRMD &emsp;&emsp;&emsp;&emsp;&emsp; (b) OR-PCA

Figure 1: Performance of subspace recovery under different rank and corruption fraction. Brighter color means better performance.

We first study the effectiveness of the algorithm, measured by the EV value of its output after the last sample, and compare it to the nuclear norm based online RPCA (OR-PCA) algorithm [8]. Specifically, we vary the intrinsic dimension $d$ from $0.02p$ to $0.5p$, with a step size $0.04p$, and the corruption fraction $\rho$ from $0.02$ to $0.5$, with a step size $0.04$. The results are reported in Figure 1 where brighter color means higher EV (hence better performance). We observe that for easier tasks (*i.e.*, when corruption and rank are low), both algorithms perform comparably. On the other hand, for more difficult cases, OMRMD outperforms OR-PCA. This is possibly because the max-norm is a tighter approximation to the matrix rank.

We next study the convergence of OMRMD by plotting the EV curve against the number of samples. Besides OR-PCA, we also add Principal Component Pursuit (PCP) [6] and an online PCA

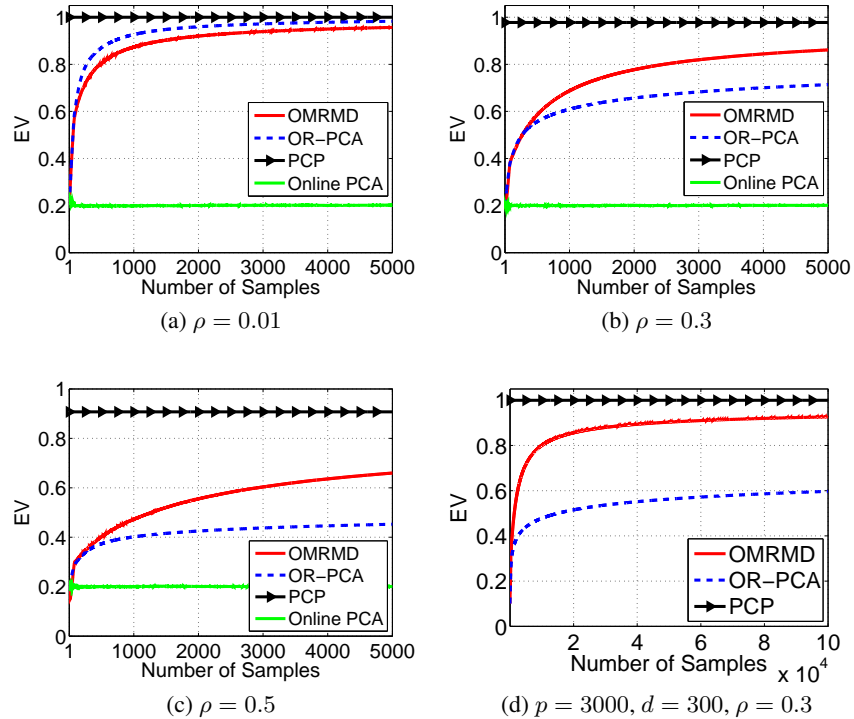

Figure 2: EV value against number of samples. $p = 400$ and $d = 80$ in (a) to (c).

algorithm [1] as baseline algorithms to compare with. The results are reported in Figure 2. As expected, PCP achieves the best performance since it is a batch method and needs to access all the data throughout the algorithm. Online PCA degrades significantly even with low corruption (Figure 2a). OMRMD is comparable to OR-PCA when the corruption is low (Figure 2a), and converges significantly faster when the corruption is high (Figure 2b and 2c). Indeed, this is true even with high dimension and as many as $100,000$ samples (Figure 2d). This observation agrees with Figure 1, and again suggests that for large corruption, max-norm may be a better fit than the nuclear norm. Additional experimental results are available in the full paper.

## 6 Conclusion

In this paper, we developed an online algorithm for *max-norm* regularized matrix decomposition problem. Using the matrix factorization form of the max-norm, we convert the original problem to a constrained one which facilitates an online implementation for solving the original problem. We established theoretical guarantees that the solutions will converge to a stationary point asymptotically. Moreover, we empirically compared our proposed algorithm with OR-PCA, which is a recently proposed online algorithm for nuclear-norm based matrix decomposition. The simulation results suggest that the proposed algorithm outperforms OR-PCA, in particular for harder task (*i.e.*, when a large fraction of entries are corrupted). Our experiments, to an extent, empirically suggest that the max-norm might be a tighter relaxation of the rank function compared to the nuclear norm.

## Acknowledgments

The research of Jie Shen and Ping Li is partially supported by NSF-DMS-1444124, NSF-III-1360971, NSF-Bigdata-1419210, ONR-N00014-13-1-0764, and AFOSR-FA9550-13-1-0137. Part of the work of Jie Shen was conducted at Shanghai Jiao Tong University. The work of Huan Xu is partially supported by the Ministry of Education of Singapore through AcRF Tier Two grant R-265-000-443-112.

## Footnotes

[1] Also known as the trace norm, the Ky-Fan $n$-norm and the Schatten 1-norm.

[2] Also known as the $\gamma_2$-norm.

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
