[Reviews · NeurIPS 2014]

Submitted by Assigned_Reviewer_42

The paper considers a problem where a matrix Z is to be decomposed into a low-max-norm matrix X, and a sparse matrix E, which is formulated as a convex problem with max-norm and L1 regularization. This can be solved using batch methods, but the focus here is on an online algorithm, which does a single pass over Z's rows.
The main trick here is to reduce the original problem to another problem, which decomposes over Z's rows, and hence can be attacked using an online learning procedure. The paper proposes an algorithm, proves that it asymptotically converges to a stationary point, and provides experiments on synthetic data.

I liked the basic idea of considering online matrix decomposition using the max-norm, which is an interesting and less-explored alternative to the nuclear norm. The algorithm derivation uses a simple but non-trivial trick, and the experiments seem promising. On the other hand:
- I had problems with understanding the motivation to the problem through matrix completion: First, in matrix completion the data matrices are very sparse, so there is no reason not to use batch methods. Second, I couldn't follow the explanation in section 1.2 why matrix completion can be cast into this framework: I understand that the equation in line (80) is equivalent to matrix completion when C -> infinity, but why is it true that the equation is equivalent to the proposed framework (equation 1.1)? I'd appreciate it if the authors can explain this in their rebuttal.
- The algorithmic derivation appears to be quite similar to OR-PCA (cited paper [8]), with the twist of using the max-norm rather than the nuclear norm, and an additional trick to get decomposability along rows. This makes this work a bit incremental.
- The analysis, although far from trivial, results in a rather weak asymptotic convergence guarantee to some stationary point. This appears to be weaker than what can be obtained for related algorithms which use other approaches, such as OR-PCA.
- The algorithm is computationally demanding: At each iteration, we need to solve a d*d matrix optimization problem, where d >= ambient data dimension to the best of my understanding (see detailed comments below). Although warm-start partially alleviates this, it's not clear if we end up with an algorithm whose runtime is much better than a batch method.

Overall, I think the paper has some interesting ideas, but also a few significant weaknesses.

Other issues/comments
--------------------
- Assumptions in section 4.1: It is stated that they can be made valid by adding a bit of regularization, but this also introduces additional terms into the optimization problem, so it's not clear if the result would still hold.. This should be clarified.
- Is it possible to use the algorithm in a batch setting, by sub-sampling individual rows uniformly at random?
- In the proposed algorithm, one needs to point out how d should be chosen. To the best of my knowledge, one needs to pick d >= max{p,n} to ensure that proposition 2.1 still holds. In that case, this issue should be explicitly discussed.

Minor comments
------------------
- The manuscript is in A4 size rather than letter size.
- line 45: "even" -> "even if"
- line 56: "...E, we aim..." - comma should be period.
- The introduction motivates the max-norm but formulation (1.1) also includes an additional E component with L1 regularization - this should be more clearly motivated (e.g. to handle sparse contamination)
- Line 88: "in literature" -> "in the literature".
- Line 173: For equation 2.7 to be true, one needs to assume that each row of the matrix Z is sampled i.i.d. from some underlying distribution. I didn't see this explicitly stated anywhere..
- Line 181: "alternative" -> "alternating"(?)
- LIne 232: Should ||r||_2 be ||r||_2^2? Otherwise the expression may not be strongly convex.
- Line 350: "groundtruth"
- Experiments: What value of d did you take?
- Line 404: "In specific" -> "specifically"
- The authors mention at several places that the max-norm is a tighter approximation of the rank than the nuclear norm. To the best of my knowledge, this is imprecise: It's a different relaxation, using a different underlying norm assumption on the latent factors (see for instance the introduction in the cited paper [13]). At most, one can argue that it appears to be a better approximation in many cases.
- It's hard to see from Figure 1 that OMRMD is better than OR-PCA: They both look more or less the same.
Summary: The paper proposes an online algorithm for max-norm-regularized matrix decomposition. It has some interesting ideas, but also a few significant weaknesses.

Submitted by Assigned_Reviewer_43

Online Optimization for Max-Norm Regularization

Summary: This paper addresses the max-norm regularized matrix decomposition problem (MRMD). The main originality of the algorithm proposed in the paper is that it is on-line. This is done by converting the max-norm regularization problem to a matrix factorization problem adapted to on-line implementation. Theoretical convergence results are provided, as well as empirical results with comparison to other state-of-the-art algorithms.

Quality: The paper is very well written and presented. In particular, the authors made an effort to outline proofs. The approach is evaluated both on the theoretical and empirical point of view. Empirical results show that it outperforms OR-PCA.

Clarity: The contribution of the paper is clearly identified and presented.

Originality: This paper brings an increment to the current literature, by offering an on-line approach to the MRMD problem.

Significance: This paper definitely brings something interesting to the field.

N.B.: I did not checked the proof of theoretical results.
Summary: This paper proposes an on-line approach to the max-norm regularized matrix decomposition problem. Both theoretical and empirical results are of interest.

Submitted by Assigned_Reviewer_44

This paper proposes an online algorithm to solve max-norm regularized problems. The asymptotical convergence of the algorithm is prooved. Some experiments are given.

Overall, I find this paper very interesting, the design of online algorithms being necessary to face streams of data. Moreover, considering low-rank matrix factorization, max-norm regularization is known to outperform the nuclear norm regularization, from both theoretical and empirical points of view.

To me, the weak point of the paper is the experimental section, though it already conveys some interesting observations.
I wonder how the algorithm behaves in a non stationary environment (slowly changing environment) since aiming at dealing with big data requires the ability to cope with such environments.
Summary: Very interesting paper, though the experimental part is rather limited.
Author Feedback
Author rebuttal: To Reviewer 42: We highly appreciate your very detailed comments. The space allowed for rebuttal will be mostly allocated to address the issues you raised

Firstly, we should clarify the notations. The matrix Z is of size p*n, i.e. n observations with p dimensions (p is the ambient dimension). d is the intrinsic dimension.

Q1: I had problems with understanding the motivation through matrix completion…

R1: In certain important practical problems, like industrial-scale recommendation systems, the data matrix can be extremely large. Although it is always sparse, it cannot be loaded in the memory for batch learning. In this case, it can be significant to derive an online implementation, which might be one concrete step to make these techniques useful and acceptable in industry. We will add more references from web conferences.

What we meant of "equivalent" is that for the MC case,it's equivalent to have a *weighted* \ell_1 norm, instead of the \ell_1 norm as in the vanilla case. However, our method and analysis easily extends there. To see this, we only need to justify that the MC problem can be solved with the proposed algorithm and our theoretical results still hold.

The main difference of MC and Eq 1.1 is the \ell_1 regularization, resulting in:
1. a new regularization on E in Eq 2.1 and 2.2: | A \circ E |_1
2. a new regularization on e for \tilde{l}(z,L,r,e) in Eq 2.4: | a \circ e |_1, where ‘a’ is a column vector
Other equations, 2.5, 2.6, 2.7 and 3.1 are defined as we did in the paper.

Then we can make a slight change of our algorithm for solving MC. In step 3 of Alg 1, given z_t and L_{t-1}, we optimize the newly defined \tilde{l}(z_t,L_{t-1},r,e) over r and e, under the constraint |r|_2 <= 1. The optimal r can be solved by Alg 2 and Alg 3 (see supplementary). Note that a_t is a vector with each entry equals to C or 1/C. We use U to denote the set of indices in a_t with value C and V for the others. Then, Eq 2.4 can be written as:
\tilde{l}(z, L, r, e) = (|z_U - L_U r_U - e_U|_2^2 + |a_U \circ e_U|_1 ) + (|z_V - L_V r_V - e_V|_2^2 + |a_V \circ e_V|_1)
= (|z_U - L_U r_U - e_U|_2^2 + C |e_U|_1) + (| z_V - L_V r_V - e_V|_2^2 + 1/C |e_V|_1)

Note that the second equation holds because entries in e_U are all with value C and those in e_V are all with 1/C. Then we can update e_U and e_V separately with soft-thresholding operator because L and r are both given. That is, in step 9 of Alg 2, the update rule for e is:
e_U = S_{C}[z_U - L_U r_U]
e_V = S_{1/C}[z_V - L_V r_V]
Now we have clarified the step 3 of Alg 1 for MC. In step 5, the \ell_1 regularized term of e has no impact on the update of L. So we can remove it. Thus, the step 4 and step 5 are same with the paper.

At last, all of our theoretical analysis hold for the MC problem which can be easily verified.

Q2: The algorithmic derivation appears to be quite similar to OR-PCA…

R2: Yes, the algorithm looks similar to [8], with two major differences:
1. [8] is formulated as an unconstrained optimization problem, where there is an equation constraint in our formulation, making our problem more complicated.
2. Compared with the smooth objective function in [8], ours is non-smooth resulted by the max-norm (the |L|_{2,\infty}^2 term in Eq 2.5 and 3.1 is non-smooth). So the theoretical analysis of ours is more difficult.

Q3: The analysis, although far from trivial,results in a rather weak asymptotic convergence…

R3: Yes, OR-PCA guarantees global optima. OR-PCA used the nuclear norm as convex surrogate, which has been widely investigated. In contrast, the max-norm is more complicated and less explored. To the best of our knowledge, there is no *practical* algorithm for max-norm regularized problems that provided global optima guarantee(SDP can’t scale up to large problems[13]). From the empirical study, we observe that the stationary point of ours usually performs better than OR-PCA.

Q4: The algorithm is computationally demanding…

R4: As we discussed above, d is the intrinsic dimension and is hence small in problems of interest. Please also note that the optimization only involves quadratic and linear terms, so it is doable for reasonable d. From our observation, our proposed algorithm enjoys a comparable running time with the batch method[6].

Q5: Assumptions in section 4.1…

R5: It can be easily checked that everything holds after adding some regularization. We will clarify it in our revised version.

Q6: Is it possible to use the algorithm in a batch setting…

R6: It is doable. But the main goal of online implementation is to mitigate memory bottleneck, so in the batch setting where the memory is not an issue, while we can still do online implementation, the benefit is not clear.

Q7: One needs to point out how d should be chosen…

R7: We state our selection of d in line 404 for Fig 1. In Fig 2, the d is chosen as 100 and the p is 400. Prop 2.1 always holds in spite of the number of d. Please check our proof in the supplementary.

Q8: Experiments: What value of d did you take
R8: Please see our response to Q7.

Q9: The authors mention at several places that the max-norm is a tighter approximation…
R9: When we refer to "tighter approximation" we mean a better generalization error bound, which was examined in [20,21] for collaborative filtering. We will modify the statement to be precise. Thank you.

Q10: It's hard to see from Fig 1 that OMRMD is better…

R10: Please kindly refer to the additional experiments in our supplementary(Fig. 3 and 4). We observe that under the low rank setting(Fig. 3), the max-norm is more robust when there is a large corruption than the nuclear norm.

--
Dear Reviewers 43 and 44: Thank you very much for the encouraging comments. We will make sure all Reviewers' concerns are adequately addressed in the revision. The future work mentioned by Reviewer 44 on studying the problem in the non-stationary environment is very interesting.